# Electrochemical Circuit Model Based State of Health Prognostics for Evaluation of Reusability of Lithium-Ion Batteries from Electric Vehicle

**DOI:** 10.3390/molecules29143325

**Published:** 2024-07-15

**Authors:** Hyunchul Kang, Minki Oh, Jaekwang Kim, Eunseon Shin, Keebum Hwang, Soyeon Kim, Youngmin Chi, Chulwan Park, Songhun Yoon

**Affiliations:** 1School of Integrative Engineering, Chung-Ang University, 84, Heukseok-ro, Dongjak-gu, Seoul 06974, Republic of Korea; guscjf763@naver.com (H.K.); dhalsrl2670@naver.com (M.O.); kjk9025@hanmail.net (J.K.); eunseon05@naver.com (E.S.); hwang1443@naver.com (K.H.); exitkid@naver.com (S.K.); youngmin511@cau.ac.kr (Y.C.); 2Department of Automobiles, Seojeong University, Seoul 05006, Republic of Korea; myriad@seojeong.ac.kr

**Keywords:** lithium-ion batteries, state of health (SOH), equivalent circuit model (ECM), electrochemistry, the used cells

## Abstract

For the purpose of predicting the state of health of already used lithium-ion batteries from 85 kWh electric vehicles, a simplified equivalent circuit model is utilized to estimate the electrochemical time constant from constant current discharge profiles. The grading process among as-obtained LIB cells is classified into three level types according to the remaining capacity and direct current resistance. Theoretically, the logarithmic equation describing cycling behavior is derived and utilized in the prediction of the state of health of the used cells. After the selection of the electrochemical time constant obtained from the best-fitting results in constant current discharge data, the suitable cycle number until the 20th cycle was selected for the prediction of the state of health after the 250th cycling data, which revealed that a narrow error range below 5% was for high and medium battery grades. Also, this error range became abruptly wider in lowest grade batteries, indicating that our proposed model for cycling behavior was highly useful in the prediction of the future state of health of the used batteries.

## 1. Introduction

In recent years, worldwide attempts to lower carbon emissions have prompted researchers to intensively investigate ways to produce and store sustainable energy [1]. However, sustainable energy is not readily available because of the imbalance between production and consumption in real-life applications resulting from production inefficiency. Hence, appropriate energy management is required to enable the efficient application of produced energy to solve the global climate crisis. A smart grid is an electrical network that enables the integration of the behavior of all users connected to the network to minimize losses during power supply. Electric vehicles (EVs) and energy storage systems (ESSs) are typical components of these smart grids, and they have emerged as breakthroughs in energy management because they provide storage stability and intercompatibility. These stable energy storage devices are largely based on lithium-ion batteries (LIBs) owing to the high availability and low self-discharge rate of these batteries. Moreover, LIBs have become more attractive to the grid industry because of their high power, high energy density, and long cycle life [2]. EVs and ESSs generally use electrical power packs/modules composed of thousands of individual LIB cells. However, LIBs are still susceptible to various aging mechanisms, which can lead to the degradation of the battery performance after repeated charge/discharge cycles [3]. Furthermore, the degradation of LIBs can result in thermal runaway within the constructed module/pack under extreme conditions. During thermal runaway, a gas is typically generated inside the cells or the temperature of the battery increases rapidly, which can cause the battery to undergo spontaneous combustion [4]. Although this does not commonly occur during LIB operation, it poses considerable safety risks for grid industries, including EVs and ESSs, as thousands of cells are located adjacent to each other in a battery pack. Occasionally, accidents involving only a few cells can lead to detonative chain reactions. Therefore, in terms of battery safety and reliability, predicting and managing the state of health (SOH) of LIBs with a limited cycle life is important [3,5]. In addition, as the global EV market expands, the consumption of batteries, which generate waste that is hazardous to the environment after they have reached their end of life (EOL), is a global concern. This is because batteries contain valuable transition metals such as Li, Co, Ni, Al, Cu, and Mn, as well as toxic organic compounds [6]. In this respect, predicting the SOH of a battery could be a measure to classify reusable waste batteries for the development of an efficient resource circulation system [7].

The SOH, a key parameter for evaluating the condition of a battery, yields the usable current capacity, as opposed to the rated capacity, and, importantly, indicates the way in which the battery performance can be evaluated in future applications [8]. A freshly manufactured LIB cell is assumed to have an SOH of 100%, and once the current capacity decreases below a predetermined threshold, it is defined as having reached the EOL state, indicating that the pack must be replaced [9]. To date, SOH estimation for LIBs has been widely studied without understanding the various electrochemical reactions that occur inside these batteries. Several methods have been proposed for the SOH estimation of LIBs, notably the direct estimation method, which includes Coulomb counting (CC), electrochemical impedance spectroscopy, the adaptive method with a Kalman filter (KF), and the application of a particle filter (PF) [4,9,10,11]

In addition, data-driven approaches, including fuzzy logic (FL), neural networks (NNs), support vector machines (SVMs), and artificial intelligence (AI), have been utilized in preliminary SOH studies [12,13,14,15]. Although various attempts have been made to use these methods meaningfully, they are mostly based on statistical calculations that do not hold any electrochemical meaning associated with the reactions inside batteries, which can exclude uncertainties such as aging, external factors, and temperature. Furthermore, existing methods are computationally complex and require large amounts of data [16], which makes their use in practical applications difficult. Also, the SOH is mainly affected by internal electrochemical processes, mechanical factors, and electrode composition, none of which can be fully reflected in these estimation methods [17]. Therefore, developing a meaningful model capable of mimicking the internal electrochemistry-based phenomena is necessary to justify the evaluation of the SOH using optimal parameters. As an alternative to these complicated approaches, the use of a simple equivalent circuit model (ECM) has been reported to describe the electrochemical reactions in LIBs [18]. Notably, the ECM is only based on the combination of linear circuit elements, such as Ohmic resistors, capacitors, an inductor, parallel constant-phase elements (CPEs), and a voltage source. This equivalent circuit is widely known as a Randles circuit. Using this model, the relationship between the voltage, current, and internal resistance can be parameterized by balancing the current. Typically, the ECM should be applied to the region in which the voltage–current characteristics have a linear relationship [19], which is difficult under normal cycling conditions. Because SOH measurements and predictions should be conducted under normal cycling conditions, a suitable governing equation for SOH prediction must be determined based on an ECM to simplify the electrochemical system.

This paper presents a simple ECM to describe the electrochemical reactions for SOH measurements. The ECM parameters can be summarized in the form of simple mathematical equations using the nonlinear relationship between the potential capacity and capacity cycle life against a basic background based on the electrochemical time constant (TC). The TC values have electrochemical implications that are characteristic of several electrochemical reactions, such as ionic transport within the bulk phase and surface film, charge transfer at the active material interface, and chemical diffusion within the particles. This can be named the TC-based SOH prediction method. The mutual TC of the simple discharge patterns determined using the basic ECM was compared with the empirically obtained cycling performance to validate the equations of our empirical model. Optimizing the SOH prediction required us to select a suitable cycle of the battery operation for analysis. This method was applied to the evaluation of LIB cells that reached their EOL after being used in EVs. These cells were classified into three groups according to their available capacity and internal resistance.

## 2. Experimental and Theory Formulation

### 2.1. Experimental

Typically, an 85 kWh battery pack for an electric bus was disassembled in order to obtain individual cells, which had a nominal capacity of 15 Ah. The disassembled individual cells were utilized to measure the available capacity and DC resistance. For the electrochemical measurement of pouch cells, their capacity was measured at the 1 C rate (15 A) for 250 cycles from 2.8 to 4 V in the first charge–discharge using a WBCS3000M2 cycler (Wonatech, Seoul, Republic of Korea). All experiments were carried out at 25 °C. For reference, the pouch cell was measured by fully discharging/charging schemes. For fully charging Li-ion batteries, the constant current (CC) protocol was used. A direct current of 15 A (1 C) was used to charge the battery for the constant current part, and the cutoff voltage was set at 4 V. After fully charging, the constant current (CC) protocol was used for fully charging. A direct current of 15 A (1 C) was used to discharge the battery, and the cutoff voltage was set at 2.8 V. Using this protocol, the battery could be discharged to reach 2.8 V. So, the batteries were operated within the voltage range from 2.8 to 4 V at 15 A (1 C) for 250 cycles. The categorization into grades A, B, and C was conducted according to the measurement of DC resistance under the 1 C rate current pulse for 10 s at 50% SOC, and, also, the measured resistance values were classified evenly with a one-third contribution from A to C grades.

### 2.2. Cycle Performance Prediction Equation Based on Equivalent Circuit Model Theory

From the above ECM, the time-dependent voltage can be expressed as follows in Figure 1:(1)$$V\left(t;I\;constant\right)=\frac{Q\left(0\right)}{C}e^{-\frac{t}{R_{2}C}}+\left(V_{0}-IR_{1}\right)-IR_{2}[1-e^{-\frac{t}{R_{2}C}}]$$

Here, *Q*(0), *C*, *V*_0_, *R*_1_, *R*_2_, and *t* are the capacity at *t* = 0, capacitance, initial voltage, respective resistances in the simple ECM within Figure 1, and the measured time, respectively [7]. As shown in this scheme, the Randal circuit of (a) can be simplified into the (b) circuit model, where the information of *C*_film_ and *C*_dl_ are included within the *C* value, while *R*_film_, *R*_CT_, and *Z*_w_ are converged into an *R*_2_ value according to the measurement frequency or time.

For mathematical simplicity, the above equation is changed as follows:(2)$$V\left(t\right)=P+Qe^{-\frac{t}{\tau_{2}}}$$

Equation (2) is utilized for nonlinear least squares fitting of the discharge curve patterns without any further modification. Due to the mathematical simplicity, Equation (2) is highly suitable for estimating the approximate values of TC, *t*_2_. From this, time t_2_ values are obtained, which can represent the overall electrochemical processes. As mentioned before, the ECM model as shown in Figure 1 should include various parameters within the Randles circuit. Strictly speaking, *R*_2_ and *C* contain such various resistance and capacity components, such as Li^+^ migration of the surface film and charge transfer at the interface and following chemical diffusion into inner particles, which is explained in Figure 1. According to the frequency or measured time domain, the typical Randles circuit becomes simplified into left circuits as shown in the literature [ECM simple model]. For more insightful understanding, such complicated components are represented by *R*_2_ and *C*. P is the total summation excluding the exponential term. So, we can express P as V0-IR_1_-IR_2_. Also, Q is the proportional factor in the exponential term; Q = Q(0)/C + IR_2_. Only the time constant is used for mutual comparison with the following empirical equation.

Equation (1) can also be transformed into cycle(*p*)-dependent available capacity (*C_a_*(*p*)) by multiplying between the measured time (*T*(*p*)) at the cut-off voltage of *V*_f_ and applied constant current (*I*) under a constant cut-off potential of (*V*_f_); *C_a_*(*p*) = *I T*(p). Remember that *T*(*p*) is the function of cycle *p* since the available discharge capacity becomes reduced with an increase in cycle number. By a simple mathematical process, the following expression about *C_a_*(*p*) is given.
(3)$$C_{a}\left(p\right)=IT\left(p\right)={IR}_{2}C\;ln\frac{Q\left(0\right)+IR_{2}C}{V_{f}C-{C^{2}V}_{0}+IR_{1}C+IR_{2}C}$$

Here, *V*_f_ is the final cut-off voltage, which can be determined externally and fixed.

From the above equation, the following three approximative concepts can be utilized in order to apply a cycle capacity of *C*_a_(*p*):

*V*_0_, *R*_2_, *C*, and *Q*(0) are dependent on the cycle number.

*R*_1_ remains constant under the condition of a less effective electrolyte solution.

t_2_ (=*R*_2_
*C*) is independent during cycle increase.

For mathematical simplicity and from an electrochemical point of view, t_2_ (=*R*_2_
*C*) is the typical time constant of an individual battery system including an electrochemical process associated with a charge transfer reaction, film migration, and chemical diffusion, which only excludes the solution resistance of *R*_1_. This parameter can be assumed to be less variant without abrupt performance change.

Hence, Equation (3) can be finally expressed considering meaningful parameters related to cycle increase:(4)$$C_{a}\left(p\right)=IT\left(p\right)=l-m\prime \ln\left[x\left(p\right)+n\prime \prime \right]$$

Here, *l*, *m*′, and *n*′ are It_2_ ln[*Q*(0) + *I* t_2_], *I* t_2_, and *I* t_2_ +*V*_f_
*C*, respectively. As elucidated, the magnitude of *V*_f_ is relatively large compared with *C* (several mF value levels). Hence, the influence of *n*′ on cycle performance is assumed to be less effective. Also, a cycle dependency parameter of *x*(*p*) is expressed; *x*(*p*) = *IR*_1_*C* − *C*^2^*V*_0_.

Here, *x*(*p*) can be expanded using the following series expansion:(5)$$x\left(p\right)=\underset{n\to \infty }{\lim}\left(\sum_{k=0}^{n}{a_{k}p}^{k}\right)=a_{0}{{+a}_{1}p}^{1}{{+a}_{2}p}^{2}+{a_{3}p}^{3}\ldots$$

For empirical conditions, coefficients *a*_k_ and *k* can be determined according to data results.

If *n* = 1 condition is employed for the initial trial, the following equation can be obtained:(6)$$C_{a}\left(p\right)=IT\left(p\right)=l-m\;ln[p+n]$$

Here, *l*, *m*, and *n* can be empirically obtained from the best-fitted results in cycle data. From the *m* value, the electrochemical time constant t_2_ can be easily estimated and compared with that from Equation (2).

Finally, the two valuable Equations of (2) and (6) can be utilized for the nonlinear least squares fitting from galvanostatic discharging patterns and cycle performance results. Also, t_2_ can be accurately determined by Equation (2) and is compared with the best-fitted values in the value of *m* containing t_2_. Through this mutual comparison process, the validity of Equation (6) is evaluated, and the most suitable cycle number of *p* is presented. After the determination of *l*, *m*, and *n* values, cycle performance is predicted after long-term cycling above 200 cycles, which can represent SOH prediction. Also, our final prediction equation, Equation (6), can be developed into a more sophisticated mathematical form if the cycle dependency of other variables is clarified theoretically.

## 3. Results and Discussion

The battery pack that was used to power an EV consisted of a pouch-type LIB of 85 kWh. The battery was disassembled and classified according to the remaining capacity and the measured direct current (DC) resistance at 50% state-of-charge (SOC). Based on this classification, the cells were divided into three grades: A, B, and C. In individual-grade cells, the remaining capacity and DC resistance comprised one-third of the population. After this process, four or five cells were randomly selected from the cells in each of these grades, and they were tested using galvanostatic charge–discharge from 2.8 to 4 V at the 1 C rate of 15 A for 250 cycles.

Figure 1 displays the CC discharge patterns of two of the grade A cells for 10 cycles. The characteristic discharge patterns did not change considerably during these cycles, indicating that these grade A cells were in a good state.

The typical results for the 1st and 10th cycles, fitted with Equation (1), are displayed in Figure 2. Despite the equation being a simple exponential equation, the results confirmed that the nonlinear least squares fitting was well conducted.

As listed in Table 1, the fitted data using the five grade A cells from the 1st to 10th cycles exhibit consistent values of *P*, *Q*, and τ_2_. This suggests that the individual values of these parameters play an important role in determining the cell performance. Because the discharge patterns of these cells remain unchanged (Figure 1), separately considering these values may be less significant. Interestingly, the electrochemical time constant τ_2_ was observed to have a value in the range of 12 to 15 s for every cell. As mentioned before, this time constant τ_2_ could represent the overall electrochemical reactions, such as ionic transport within the surface film, charge transfer at the electrode interface, and chemical diffusion within bulk particles, because the employed ECM was based on the Randles circuit. Along this line, the consistent τ_2_ values imply that the overall time domain of the electrochemical reaction was approximately 10 s.

Figure 3 displays the typical galvanostatic charge–discharge cycling performance of a grade A cell until the 250th cycle. The stability of the cycling pattern resembles that of a fresh cell. This indicates that LIB cells can be used for other applications after their EOL. In addition, the expected cycling performance calculated using Equation (6) is plotted as a black line, where nonlinear least squares fitting was applied to measured data from the first 20 cycles.

To ensure the selection of a suitable cycle for the best fitting, the cycle number *p* in Equation (6) was selected as 10, 15, 20, and 250, as listed in Table 2. As shown, the expected time constant changed from 12 to 53 s for the grade A cell Ch1. Among these data, cycle number 20 exhibited a value of 18 s, similar to the fitted results (12–15 s) in Table 1, indicating that the application of Equation (6) for the prediction of the SOH is highly reasonable when using the performance data of the 20th cycle. Because the extent of change in τ_2_ was assumed to be less, despite the variation in the values of *l*, *m*, and *n* as cycling proceeded, the optimal selection of the cycle number was crucial when employing Equation (4) for the SOH prediction. Hence, the values of *l*, *m*, and *n* in Table 2 were used in the fitted results shown in Figure 3. In addition, the errors estimated from the difference between the real and expected capacity values are plotted in Figure 3b. As shown, the error range is below 3%, indicating the high accuracy of the SOH prediction using Equation (6) based on data from the 20th cycle.

A similar process was employed for another grade A cell, as shown in Figure 4 and Figure 5. The high accuracy, with an error below 4%, was maintained for another four cells, indicating the validity of Equation (6) for processing the performance data of the 20th cycle. A rough estimation of the performance data of 20 cycles at a rate of 1C would require measurements to be acquired for approximately two days. Therefore, our approach requires further improvement before it can be practically applied to predict the SOH. Furthermore, the initial capacity was measured to be in the range of 14.0 to 14.8 Ah with a smaller discrepancy.

A similar process was used to evaluate Equations (2) and (6) for the grade B and C cells. In Figure 6, a typical discharge pattern is plotted for ten cycles, and the fitted results are shown in Figure 7.

As listed in Table 3, the τ_2_ values remain constant, irrespective of the measured cells, from 14 to 16 s, implying a slight increase in the time constant for grade A cells. To utilize Equation (4), the cycle number was selected, and the results are listed in Table 4.

As shown in Table 4, the τ_2_ values range from 13 to 23 s, which constitutes a wider distribution when compared with that of the grade A cells.

This comparison reveals that grade B cells were degraded to a larger extent, based on their higher τ_2_ values with wider distributions, indicating a more sluggish electrochemical process. The cycling performance and expected capacity values are plotted in Figure 8. First, the initial capacity was measured from 14.7 to 13.5 Ah, indicating increased discrepancy. However, the predictions using Equation (6) until the 20th cycle demonstrated that the error level was as low as 4%, which is similar to that of grade A cells. Although cell degradation proceeded to a greater extent in grade B cells, prediction of the SOH was expected to be possible by selecting a suitable number of cycles.

A similar analysis was performed for grade C cells, and the results are plotted in Figure 8, Figure 9 and Figure 10. In Figure 9, a peculiar linear decrease in the discharge capacity, which differs from that of the grade A and B cells, is observed for the two cells.

As is evident from Table 5, the time constants of the discharge patterns obtained using Equation (2) are widely distributed, and the values for only two cells, Ch3 and Ch5, are detected at approximately 15 s. Because a simple classification was employed in our experiment, the three grades based on capacity and DC resistance are arbitrary. The cells for Ch3 and Ch5 exhibited high resistance but medium capacity. In addition, the ECM employed to describe the total electrochemical behavior is highly simplified.

As shown in Table 6, a suitable cycle number was selected using Equation (4). A large fluctuation in τ_2_ was observed for cells Ch 1, 2, and 4, whereas values of 28 and 10 s were obtained for cells Ch3 and 5, respectively. Based on this result, the assumption that the time constant changed less because of abrupt changes in the overall electrochemical reactions was shown to be invalid. This indicates that Equation (4) is unsuitable for predicting the SOH of highly degraded LIBs.

The estimated t_2_ values from best-fit discharge patterns are plotted in Figure 10, and the error range is also shown. As shown, TC values from grade A have the lowest error range, located within a highly accurate range of values, indicative of the consistency of our approach and a highly reasonable prediction of the SOH of grade A cells. Grade B cells have an increased error range of t_2_, and there is a wide discrepancy in the error range for grade C-type cells. Hence, it is highly valuable to utilize the TC of individual cells as the critical parameter, which can be extended into the prediction of cycling performance from a reasonable selection of SOHs based on Equation (6).

## 4. Conclusions

In our approach, a simplified linear circuit model describing LIBs was utilized in order to estimate the electrochemical time constant from constant current discharge profiles. From the derivation of a simple mathematical equation describing cycling performance, it was confirmed that the theoretical logarithmic equation was derived and highly useful in the prediction of the state of health of three grades of batteries, which were obtained from already used LIBs from electric vehicles. Using the electrochemical time constant obtained from the best-fitting results from constant current discharge data, the suitable cycle number was selected for the prediction of the state of health in cycling data until the 20th cycle, which revealed that a narrow error range below 5% was achievable for data on 250 cycles. Also, this error range became abruptly wider in low-grade batteries from electric vehicles.

## Data Availability

Data are contained within the article.

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
