# Peer review of "Electrochemical Circuit Model Based State of Health Prognostics for Evaluation of Reusability of Lithium-Ion Batteries from Electric Vehicle"

_molecules, 2024, doi:10.3390/molecules29143325_

Round 1
Reviewer 1 Report
Comments and Suggestions for Authors
The authors predicated the state of health in lithium batteries with the simplified equivalent model. However, there are still some doubts in this manuscript.
1. Please provide the classification of grade-A, B, and C cells. Which stage was selected for these grade-A, B, and C cells. And the authors should exhibit the related TC region of the grade-A, B, and C cells according to the equivalent model.
2. Actually, the relationship should be established by the statistical data of a large number of grade-A, B and C cells. In this work, the sample number of each grade cells is still insufficient.
3. The authors mainly introduced the simplified equivalent model of the lithium battery, but the accuracy verification of the model should be considered.
4. The units in these tables should be provided. And the representation of P in Eq. 2 should be expressed.
5. The formats in this manuscript should be unified. The “τ2” is missed in the main text, please correct.
Author Response
The authors predicated the state of health in lithium batteries with the simplified equivalent model. However, there are still some doubts in this manuscript.
- Please provide the classification of grade-A, B, and C cells. Which stage was selected for these grade-A, B, and C cells. And the authors should exhibit the related TC region of the grade-A, B, and C cells according to the equivalent model.
Answer) Thank you for your helpful comments on our work. This is a very useful question. The categorization into grades A, B, and C was conducted according to the measurement of DC resistance under 1 C rate current pulse during 10 second at 50% SOC and also, the measured resistance values were classified with evenly one third contribution from A to C grades
- Actually, the relationship should be established by the statistical data of a large number of grade-A, B and C cells. In this work, the sample number of each grade cells is still insufficient.
Answer) As you commented, more statistical data are required for the improved treatment. However, the EV cells are not highly available due to their size and capacity. Furthermore, the high current cyclers are required, which made us difficult to obtain more cells data under our lab scale. Please understand this situation.
- The authors mainly introduced the simplified equivalent model of the lithium battery, but the accuracy verification of the model should be considered.
Answer) As mentioned in our submitted manuscript, our primary research goal is to employ the new model to predict SOH using the time constant, which has a novelty in our opinion. Also, we are prepared more advanced equivalent circuit model, which will be addressed later.
- The units in these tables should be provided. And the representation of P in Eq. 2 should be expressed.
Answer) As seen in Eq. 2, P is the all of summation excluding the exponential term. So, we can express the P is V0-IR1-IR2. This is added into the manuscript according to your kind comments. Also Q is the proportional factor in the exponential term; Q = Q(0)/C + IR2.
- The formats in this manuscript should be unified. The “τ2” is missed in the main text, please correct.
Answer) We feel sorry for our error and τ2 has been revised to τ1.
I have additionally attached the revised manuscript. I greatly appreciate your help
Reviewer 2 Report
Comments and Suggestions for Authors
This research on predicting battery State of Health is nicely presented. The introduction is well-written and comprehensive, demonstrating a meaningful effort with potential impacts on battery recycling, which bring economic value and environmental conservation.
To enhance clarity, it would be beneficial to present the calculated R and C values in a table format (maybe put in SI). Additionally, the labeling of tables using "C1-C10" can be confusing, especially considering that "C" also denotes capacitance and battery grade C.
Furthermore, the significant digits in the P, Q, and τ2 values are excessive, considering the approximate nature of the model.
The author should provide more explanation on how the modeling aids in evaluating battery health, including the necessary data collection procedures and the predictions that can be made. To me, the description sounds like a requirement of 20 cycles of charge-discharge curve data at certain parameter to predict 250 cycles, but please clearify - running 20 cycles takes quite some time, is it possible to predict with less data?
Additionally, clarification is needed regarding how batteries are graded into categories A, B, and C. It's intriguing why the C grade deviates from the predicted outcomes; the author should delve into more detail to explain this phenomenon – would that requires a more complicated model? Does C grade battery have something special and doesn’t fit the model?
Author Response
This research on predicting battery State of Health is nicely presented. The introduction is well-written and comprehensive, demonstrating a meaningful effort with potential impacts on battery recycling, which bring economic value and environmental conservation.
Answer) Thank you for your kind and helpful comments on our manuscript.
- To enhance clarity, it would be beneficial to present the calculated R and C values in a table format (maybe put in SI). Additionally, the labeling of tables using "C1-C10" can be confusing, especially considering that "C" also denotes capacitance and battery grade C.
Answer) Thank you for your helpful comments on our study. This is a very useful question. The meaning of C is the cycle number, and it has been renamed as CYC. As seen in Eq. 2, P is the all of summation excluding the exponential term. So, we can express the P is V0-IR1-IR2. This is added into the manuscript according to your kind comments. Also Q is the proportional factor in the exponential term; Q = Q(0)/C + IR2.
- Furthermore, the significant digits in the P, Q, and τ2 values are excessive, considering the approximate nature of the model.
Answer) The effective number of these values are changed into the useful significant four digits. And Q and P are denoted as V for Voltage. And τ1 is denoted as s for second. We feel sorry for our error and τ2 has been revised to τ1.
- The author should provide more explanation on how the modeling aids in evaluating battery health, including the necessary data collection procedures and the predictions that can be made. To me, the description sounds like a requirement of 20 cycles of charge-discharge curve data at certain parameter to predict 250 cycles, but please clearify - running 20 cycles takes quite some time, is it possible to predict with less data?
Answer) As you mentioned, the 20 cycles seem not optimized cycles for more accurate evaluation of time constant. However, please remember these values should be similar to time constant from galvanostatic charge-discharge data. So, we employed 20 cycles data instead of other cycles. Later we will conduct more reduced cycle numbers for minimization of prediction time of SOH.
- Additionally, clarification is needed regarding how batteries are graded into categories A, B, and C. It's intriguing why the C grade deviates from the predicted outcomes; the author should delve into more detail to explain this phenomenon – would that requires a more complicated model? Does C grade battery have something special and doesn’t fit the model?
Answer) The categorization into grades A, B, and C was conducted according to the measurement of DC resistance under 1 C rate current pulse during 10 second at 50% SOC and also, the measured resistance values were classified with evenly one third contribution from A to C grades
I have additionally attached the revised manuscript. I greatly appreciate your help
Round 2
Reviewer 1 Report
Comments and Suggestions for Authors The authors have been revised this manuscript. However, there are still some errors in this version. For example, the “τ2” is still missed in the main text. Please check Page 6, 7, and 10. "Eqs (2) and 6" in Page 9. Whether the "t2" in equations is “τ2”? Please check this manuscript.
Author Response
Thank you for identifying the errors in the manuscript. All instances of τ2 have been replaced with τ1. The errors in Eqs (2) and 6, where τ2 was written, have also been corrected to τ1.